# Burst Release from In Situ Forming PLGA-Based Implants: 12 Effectors and Ways of Correction

**DOI:** 10.3390/pharmaceutics16010115

**Published:** 2024-01-16

**Authors:** Elena O. Bakhrushina, Polina S. Sakharova, Polina D. Konogorova, Victor S. Pyzhov, Svetlana I. Kosenkova, Alexander I. Bardakov, Irina M. Zubareva, Ivan I. Krasnyuk, Ivan I. Krasnyuk

**Affiliations:** Department of Pharmaceutical Technology, A.P. Nelyubin Institute of Pharmacy, I.M. Sechenov First Moscow State Medical University (Sechenov University), Moscow 119048, Russia; bakhrushina_e_o@staff.sechenov.ru (E.O.B.); sakharova_p_s@student.sechenov.ru (P.S.S.); konogorova_p_d@student.sechenov.ru (P.D.K.); kosenkova_s_i@staff.sechenov.ru (S.I.K.); bardakov_a_i@staff.sechenov.ru (A.I.B.); kashlikova_i_m@staff.sechenov.ru (I.M.Z.); krasnyuk_i_i@staff.sechenov.ru (I.I.K.); krasnyuk_i_i_1@staff.sechenov.ru (I.I.K.J.)

**Keywords:** in situ, in situ forming implants, burst release, PLGA, phase inversion, cumulative release

## Abstract

In modern pharmaceutical technology, modified-release dosage forms, such as in situ formed implants, are gaining rapidly in popularity. These dosage forms are created based on a configurable matrix consisting of phase-sensitive polymers capable of biodegradation, a hydrophilic solvent, and the active substance suspended or dissolved in it. The most used phase-sensitive implants are based on a biocompatible and biodegradable polymer, poly(DL-lactide-co-glycolide) (PLGA). Objective: This systematic review examines the reasons for the phenomenon of active ingredient “burst” release, which is a major drawback of PLGA-based in situ formed implants, and the likely ways to correct this phenomenon to improve the quality of in situ formed implants with a poly(DL-lactide-co-glycolide) matrix. Data sources: Actual and relevant publications in PubMed and Google Scholar databases were studied. Study selection: The concept of the review was based on the theory developed during literature analysis of 12 effectors on burst release from in situ forming implants based on PLGA. Only those studies that sufficiently fully disclosed one or another component of the theory were included. Results: The analysis resulted in development of a systematic approach called the “12 Factor System”, which considers various constant and variable, endogenous and exogenous factors that can influence the nature of ‘burst release’ of active ingredients from PLGA polymer-based in situ formed implants. These factors include matrix porosity, polymer swelling, LA:GA ratio, PLGA end groups, polymer molecular weight, active ingredient structure, polymer concentration, polymer loading with active ingredients, polymer combination, use of co-solvents, addition of excipients, and change of dissolution conditions. This review also considered different types of kinetics of active ingredient release from in situ formed implants and the possibility of using the “burst release” phenomenon to modify the active ingredient release profile at the site of application of this dosage form.

## 1. Introduction

Among parenteral controlled-release dosage forms, in situ forming implants (ISFI) are gaining popularity as an attractive alternative to ready-made implants. Their main advantages are controlled drug release in time and place, simple manufacturing process, and minimal invasiveness [1].

ISFIs are usually a customisable matrix that includes a hydrophilic solvent, a biocompatible and biodegradable polymer, and a suspended or dissolved active ingredient (AI) [2]. When the prepared liquid formulation is injected into an aqueous environment, phase inversion occurs—replacing the organic solvent inside the ISFI with the biological fluid surrounding the implantation site, resulting in the formation of a biphasic porous solid depot capable of releasing the drug over a long period of time [3,4].

The most used phase-sensitive polymer for drug loading is poly(DL-lactide-co-glycolide) (PLGA). Its wide application in the delivery of active ingredients in tissue engineering, dentistry, oncology, and surgery has been favoured by the properties of the polymer, such as safety, absence of toxic effects on the body, mechanical strength, biocompatibility, and degradability [5]. PLGA is already present in FDA-approved injectable products—Atridox^®^, Atrigel^®^, and Eligard^®^ systems [6].

However, one of the most significant disadvantages of PLGA-based ISFIs is the high concentrations of the released substance immediately after injection—the so-called “burst release” phenomenon [7].

In this regard, one of the most important challenges in pharmaceutical development of ISFIs today is to find a solution for levelling the initial burst release—however, the solutions proposed in various publications are not systematised and at times lack theoretical justification.

In situ systems produced by phase inversion (phase-sensitive) are rapidly developing nowadays, which can be seen by the growing research interest in them according to PubMed, the largest international database of medical publications. To increase the number of effective developments of such systems reaching the stages of clinical trials, a reasonable methodology of their development is required, including, of course, a systematic approach to overcoming “burst release”.

One of the first systematic review papers on the topic is Allison SD’s review, published in 2008, where the mechanisms underlying the phenomenon of “burst release” of AIs from PLGA microparticles were discussed [8]. When analysing the scientific literature published between 2004 and 2008, two strategies for controlling of burst release were observed: improving the wettability of AIs and changing the production technology.

The most recent significant work in this field can be considered the review by Yoo J and Won YY 2020, also focused on the phenomenon of “burst release” from microparticles [9]. In addition to the actualised strategies for overcoming the “burst”, a detailed discussion of the release kinetics and criteria for attributing the nature of the release of AIs to the “burst” release were offered in the publication.

However, it should be stated that all published reviews on the topic concern only the problem of “burst” release from PLGA microparticles, which can be interpreted to the practice of in situ system development but does not consider the peculiarities of ISFI. Thus, the aim of this review was to systematise the information on practical stabilisation of the active substances release character from PLGA-based ISFIs.

## 2. Base PLGA Characteristics

Polylactic-co-glycolic acid is a linear aliphatic copolymer obtained through block copolymerisation of its constituent monomers, lactic acid (LA) and glycolic acid (GA), taken in various ratios. It can be synthesised with any ratio of LA to GA, and the molecular weight (Mw) is in a wide range from less than 10,000 to 200,000 g/mol (Da). Its formula is demonstrated in Figure 1.

The LA:GA ratio determines the hydrophilic–lipophilic properties of the polymer. Thus, when the lactide content in the final polymer increases, its hydrophobicity increases and its solubility in polar solvents decreases, and the opposite is true when the proportion of glycolide in the block copolymer increases [10]. Also, the HLB of the block copolymer will directly affect its behaviour under physiological conditions. Thus, although PLGA itself is not soluble in water, being a hydrophobic molecule, it has the ability to swell and soften in a humid environment. The effect of water on the polymer leads to a decrease in the glass transition temperature of the polymer and changes in the polymer matrix, and consequently, in the release kinetics [11].

PLGA is degraded by hydrolysis of its ester bonds through bulk or heterogeneous erosion in an aqueous medium. Four stages of its degradation can be described in detail (Figure 2):Hydration: water penetrates the amorphous region and destroys the van der Waals interaction forces and hydrogen bonds, causing a decrease in the glass transition temperature;Initial degradation: breakage of covalent bonds with a decrease in molecular weight;Permanent degradation: carboxyl end groups autocatalyze the degradation process, and mass loss begins due to massive breakage of main chain covalent bonds, resulting in loss of integrity;Solubilization: the fragments are further degraded to aqueous soluble molecules [12].

**Figure 2 pharmaceutics-16-00115-f002:**
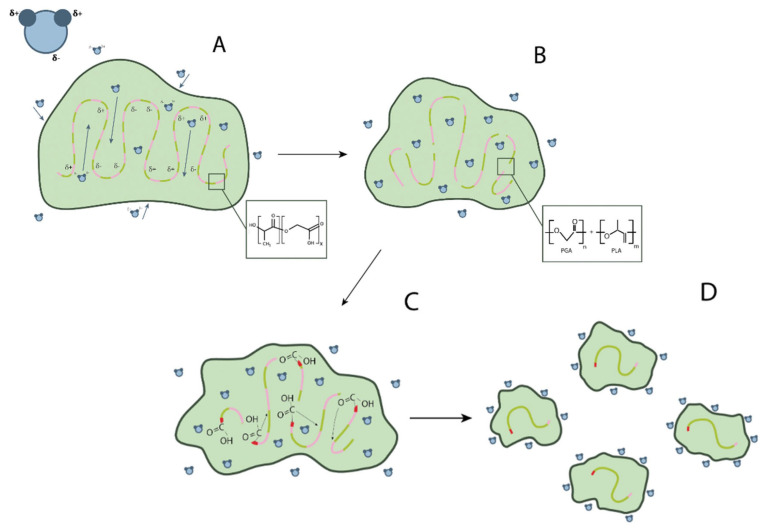
Scheme of PLGA degradation: (**A**) hydration; (**B**) initial degradation; (**C**) permanent degradation; (**D**) solubilization.

After degradation, LA and GA are formed as end products. Different parameters can influence the degradation rate:Molecular weight: when the molecular weight of conventional PLGA is increased from 10–20 to 100 kDa, the degradation rate is reported to range from several weeks to several months;GA to LA ratio: PLGAs with higher LA content are less hydrophilic, absorb less water, and subsequently degrade more slowly due to the presence of methyl side groups in PLA, making it more hydrophobic than PGA. The exception to this rule is the 50:50 copolymer, which degrades faster [13];Stereochemistry: mixtures of lactic acid monomers D and L are most commonly used to make PLGA because the rate of water penetration is higher in the amorphous regions of D and L, leading to accelerated degradation of poly-lactic-co-glycolic acid [13];Modification of end groups: polymers that are capped by esters (as opposed to free carboxylic acid) have longer half-lives [14]. Moreover, the shape of the polymer chain strongly influences the degradation behaviour of PLGA depending on water availability. In addition, low pH accelerates the degradation of PLGA due to the autocatalysis of the reaction [15].

When PLGA is used as a matrix for drug delivery, manufacturers often only specify the amount of lactide to glycolide (LA/GA) ratio of PLGA, but do not provide the molecular weight, monomer sequence, end groups [16], polymer structure (linear or star-shaped chain branching), crystallinity, glass transition temperature, surface morphology, hydration rate, drug content, etc. [17]

The development of new drugs and replication of successful experiences in the development of PLGA-based delivery systems is complicated by the lack of standardized protocols for this polymer [18]. The polymer structure can be variable due to the different reactivity of the two constituent monomers [19] and their random distribution, which makes it difficult to obtain samples with the same structure. The properties of PLGA may vary from manufacturer to manufacturer due to different approaches to the synthesis method and processing conditions that will affect the characteristics of the end polymer.

## 3. Possible Causes of the “Burst Release” Problem

The release of the active ingredient into the surrounding tissues occurs in several steps: diffusion of the AI, swelling of the polymer, degradation and erosion of the polymer, pore formation and AI–polymer interaction [20].

First, the solvent diffuses into the liquid environment from the implant through the microporous structure (Figure 3). This occurs until the polymer begins to degrade. Finally, because of biodegradation, there is a period of enhanced release that promotes “bulk erosion” of the carrier matrix [21].

There are many theories to explain the possible reasons for the burst release from a monolithic system, but drugs with different physicochemical properties have different release kinetics. Depending on the intended use of the active ingredient, the burst release effect can be either a way to achieve high concentrations at the injection site for immediate pharmacological action or a side effect leading to complications. Usually, “burst” is observed in low-molecular-weight drugs due to small molecular size and osmotic pressure which increases the concentration gradient [22]. Burst release is also characteristic of osmotic systems—in case of deformation of their membranes [23].

As early as 1997, Batycky R.P. et al., who studied in situ systems based on PLGA, suggested that during the manufacturing process, after solidification with the formation of a solid matrix, part of the active substance is retained on the surface. The effect was particularly pronounced in the case of high loading of the polymer by the drug, with subsequent release at the site of administration immediately after activation. Hydrophilic molecules were initially localized near occlusions (“macropores”) and then diffused by convection together with water during drying or storage, when water moved from the centre to the surface and evaporated [24].

Joiner JB et al., in 2022, conducted a study to investigate the in vitro release of various low-molecular-weight compounds with different logP and pKa values from PLGA-based ISFI. The data obtained were compared with polymer degradation measured by lactic acid release and PLGA molecular weight change. Taken together, these results demonstrated that hydrophilic substances have a higher release rate within 24 h (22.8–68.4%) and complete release in 60 days, while hydrophobic molecules have a lower rate (1.8–18.9%) and can maintain drug release from 60 to 285 days [25].

Thus, it has been shown in several studies that the value of the release “burst” and the sedimentation rate depend on many factors. Based on the analysed data, we proposed a system of “twelve factors” concerning both the matrix formulator, AI release, and other matrix components affecting the phenomenon of burst release (Figure 4).

Among the above list of effectors, factors 1 to 6 can be emphasized as those that cannot be modified during the pharmaceutical development process, and their influence must be considered when “burst” release correction is required. Factors 7 to 12 are controllable—in other words, they can be altered in the investigation of the optimal composition of the phase-sensitive matrix.

### 3.1. Porosity

In the work of Matsumoto A. et al., it was found that pores in the polymer matrix are formed both because of water absorption and in the process of the polymer degradation/erosion [26]. These processes are influenced by many factors, such as the properties of PLGA, the addition of plasticisers and salts, the acidity of the degradation environment, etc. [27,28]. An important role is played by acid catalysis, the sources of which can be either low pH of the environment due to the addition of acids (non-autocatalytic effect) or carboxyl groups of polymer chains (autocatalytic effect) [29,30]. PLGA is a polymer that can undergo hydrolytic degradation into low-molecular-weight hydroxy acids. Any PLGA-based matrices are bulk-erodible and can form acids that further diffuse into the media or body fluids where they will be neutralised. However, in polymer implants, mass transfer is rather slow, so the rate of acid formation can be much higher than the rate of neutralisation. As a result, micro-pH values in the system decrease [31,32] and polymer degradation is accelerated. This can interfere with the stability of AIs of protein nature or lead to accelerated release of AIs due to increased mobility of molecules and decreased average molecular weight [33].

In order to reduce the effects of in situ PLGA-based systems autocatalytic degradation, various strategies can be used. One of those is to lengthen the diffusion pathways of acids and bases by increasing the matrix size [34] or through the addition of bases to the ISFI composition [35]. Furthermore, increasing the mobility of diffusing molecules can also help to prevent autocatalytic effects. For this purpose, the porosity of the polymer matrices can be increased to accelerate the diffusion of water-soluble molecules. This can be achieved using appropriate preparation techniques, such as extraction or artificial pore creation using multiple emulsion technology [33].

In the study by D. Klose, the main aim was to obtain highly porous drug-loaded and drug-free PLGA-based microparticles of different sizes. The effects of porosity and microparticle size on drug release kinetics and degradation behaviour of the polymer were studied. The results showed that the release rate of lidocaine was almost unchanged with increasing microparticle size and decreasing drug concentration gradient. However, in porous PLGA-based microparticles, water-soluble acids and bases diffuse much faster than in non-porous microparticles [29].

Yuan P. et al. also showed the influence of the pore type formed by multiple microemulsions in PLGA-based microspheres on the release of both small molecules and proteins (fluorescein isothiocyanate labelled BSA (FITC-BSA), rhodamine B, interleukin-4 (IL-4), melatonin) [36]. Using different methods to produce water/oil/water type emulsions, samples with both pores only on the particle surfaces and a combination of internal and surface pores were obtained. These two types of porous microspheres exhibited different release kinetics of AIs of both protein and synthetic nature. The release of AI from microspheres with only external pores was significantly faster (up to 5 times) than from microspheres with both external and internal pores, which proves other hypotheses about the influence of not only the number or presence of pores but also the morphology of the whole structure.

### 3.2. Swelling of the Polymer

The role of PLGA swelling is often neglected in the literature when explaining models of AI release from drugs [37]. However, studies show that PLGA swelling can have an important effect on the process of AI release—for instance, due to the increase in polymer size, micropores can be blocked, which leads to a decrease in drug diffusion [38]. Gasmi H. et al. studied the swelling of individual PLGA particles using optical microscopy and noted the relationship between swelling and the release of acidic substances from different types of particles [38].

In numerous papers of the research group of College of Pharmacy, Univ, Lille (France), it was noted that at the initial stages, the amount of water that can enter the PLGA-based system is limited due to hydrophobicity and spatial entanglement of macromolecules. The solvent penetrating into the implant starts to hydrolyse polyester bonds; as a result, new -COOH end groups are formed, which reduces the hydrophobicity of the molecule and its spatial entanglement. The osmotic pressure of the system increases, which leads to a significant increase in water influx into the implant. It is observed that during swelling, the volume of the system can increase up to 1700% and the water content exceeds 90%, which favours significant dissolution of the active ingredients. Only after that do AI molecules become sufficiently mobile for diffusion, which eventually leads to their release [38].

Thus, the swelling of implants and microparticles based on PLGA is a limiting factor in the initiation of AI release, allowing delayed release of AIs, which leads to changes in dissolution and diffusion conditions. It should be noted that other types of implants can lead to additional mass transfer processes, and no burst release was found in the studied originally non-porous implants [38].

### 3.3. LA/GA Ratio

As mentioned above, the molar ratio and arrangement of lactic and glycolic acids determine the physicochemical characteristics of PLGA. The bigger the ratio of lactide to glycolide, the more hydrophobic the substance becomes—due to the presence of a side methyl group in lactic acid [39]. Copolymerisation of glycolide with D,L-lactide in different ratios yields products with different biodegradation rates. The biodegradation time of polylactide ranges from 12 to 16 months, but it can be accelerated by increasing the proportion of glycolide in the composition. The polymer degradation time is shorter for more hydrophilic polymers with lower molecular weight, higher amorphous part, and higher glycolide content in copolymers.

The results of physical and mechanical tests of polymer matrices of PLGA compositions indicate that reducing the content of D,L-lactide in the compositions by 10% affects the material durability, which decreases by 34%. And the increase in the number of lactide fragments increases hydrophobicity but decreases crystallinity.

Simultaneously, an increase in the proportion of glycolide in the copolymer leads to a significant increase in elasticity, and the maximum elasticity index of the polymer was observed for PLGA 60/40 samples.

In the work of Nasonova M.V. et al., it was shown that there are statistically significant differences in elasticity for samples PLGA 70/30 (*p* = 0.048) and PLGA 60/40 (*p* = 0.019). By varying the amount and time of monomer addition, the fine structure of the material and its properties, such as the time of complete polymer degradation, can be changed [40].

The ratio of glycolic acid to lactic acid in PLGA-based ISFI can affect the implant formation rate and AI release profile.

Studies indicate that PLGA with a 50/50 ratio of lactide to glycolide shows the fastest biodegradation rate, within 50–60 days. Due to the hydrophilic nature of glycolic acid, a PLGA composition with higher proportion of glycolic acid has a higher rate of hydration, while a PLGA composition with a lower proportion of glycolic acid results in slow drug release [10].

Vey E. et al. described a degradation study of four polymers with different LA/GA ratios: 50/50, 65/35, 75/25, and 95/5. The study showed that PLGA 95/5 lost only 3% of its mass in 10 days, while PLGA 50/50 lost 20%. However, after 30 days, the decomposition process in PLGA 95/5 was significantly accelerated. According to the researchers, this can be explained by the fact that LA-chains are more hydrophobic, which slows down water diffusion and leads to slower kinetics of implant degradation. Compositions calculated from infrared and Raman spectra showed that the percentage of lactic and glycolic acids remained approximately constant up to 10 or 15 days for different LA/GA ratios. As degradation proceeds, the percentage of lactate sites in the compound increases relative to the glycolic sites. The degradation rate constant of the glycol unit in all copolymers is 1.3 times higher than that for the lactic unit, and the degradation rate constants between the LA and GA links decrease with increasing initial lactic acid content in the copolymer. These results suggest that water diffusion in the sample has a significant effect on the degradation kinetics of copolymers [41].

Our recent studies [42] investigated the effect of the ratio of lactide to glycolide in PLGA on the rate of implant formation and the diffusion of water-soluble colourant from the implant in phosphate buffer environment and in a biorelevant in vitro model of gingival soft tissue [43]. In a preliminary implant formation rate test for the NMP-PLGA 75/25 systems, phase inversion and implant formation took about 3 s, which was visually instantaneous, while the NMP-PLGA 50/50 formulation took about 5 min and gentle shaking of the test tube to form an insoluble matrix. It was found that in the cavity of the artificial agar tooth socket, the system NMP-PLGA 75/25 already forms a matrix insoluble in water after 1 h, and the implant of the composition NMP-PLGA 50/50 does not solidify for more than 3 h. At the same time, the diffusion of the colourant into agar blocks, the density of which corresponds to physiological parameters of soft tissues, for the implant NMP-PLGA 75/25 was 1060 µL, and for NMP-PLGA 50/50 was 641 µL.

### 3.4. Impact of End Groups

As discussed in Section 3.2, the carboxyl groups formed during the swelling of PLGA increase the hydrophilicity of the molecule [37].

In the research, it has been shown that formulations made from PLGA polymers with carboxyl groups have different properties from those made from polymers with similar molar weight (MW) of PLGA but with ester end groups. For example, they absorb more water, have higher polymer mass loss, and accelerate the release of hydrophobic substances [44]. AI release of hydrophilic AIs is also influenced by the factor of end groups. The work of Wang J et al. is devoted to the development of PLGA microspheres loaded with hydrophilic doxycycline hyclate [45]. PLGA-based microspheres with acidic end groups showed lower encapsulation efficiency and higher doxycycline release rate.

It should be noted that the effect of these end groups on AI release decreases with increasing molecular weight and polymer chain length, which increases the hydrophobicity and decreases the total number of end groups. Therefore, the influence of hydrophilic end groups in long-chain polymers decreases as the MW of the polymer increases. The concentration at which the end functional group no longer affects controlled drug delivery is a matter of discussion [46].

### 3.5. Molecular Weight of the Polymer

The molecular weight of PLGA, as a factor affecting the kinetics of AI release and degradation of the co-polymer, is considered by many researchers together with the nature of the end groups. It is known that the degradation rate of PLGA depends on its MW, respectively on its viscosity, since short chains degrade faster than long chains [47]. In a study by Patel RB et al., the highest drug release rates were obtained from low-molecular-weight PLGA [48], which had the densest surface concentration of acid end groups. An intermediate drug release profile was obtained using a mixture of high- and low-molecular-weight PLGA polymers [44].

Yewey GL et al. [49] reported a different relationship between protein release and the MW of the polymer. They investigated two PLGA formulations with the same polymer concentration but different molecular weight, which were mixed with an equivalent amount of myoglobin. Initially, the two formulations released the protein at the same rate, but over time, the lower-molecular-weight polymer released about 10% more myoglobin than its high-molecular-weight analogue. The burst nature of the release was not reported in this work.

Luan X. and Bodmeier R. [50] have reported that ISFI made from a low-molecular-weight polymer resulted in a much lower initial release than a medium weight polymer. From these findings, it can be concluded that the relationship between polymer molecular weight and ISFI release is controversial and requires further investigation.

In another experiment, the effect of polymer molecular weight on the release of a model peptide, leuprolide acetate, from ISFI was studied. PLGA 50:50 copolymers with MWs of 12 kDa, 34 kDa, and 48 kDa were taken as materials. The surfaces of the three solidified membranes were examined using micrographs. The low- and high-MW compositions had smooth porous surfaces, but the composition with PLGA having lower MW showed higher porosity and larger pore diameter. The medium-MW implant had a different structure from the two previous compositions: it appears as a cellular surface. Characterisation of this particular structure shows surface pores with a polygonal pattern that are embedded in the upper surface of the membrane. According to the authors, the “burst” release of the 34 kDa MW composition is due to the cracking of microstructures or cells that may contain dissolved AI [51].

### 3.6. Structure of the Active Molecule

Despite the importance of all listed factors that can influence the course of the processes occurring with the in situ implant after its introduction into the organism, the stereochemical structure of the active substance also appeared to be important. The study of Simo et al. [52] showed that four polymeric delivery systems of methacrylic derivatives of ibuprofen release its enantiomers with a small excess of S-enantiomer.

In the study by Wang S. et al., the authors investigated how S- and R-isomers are released from racemic polymeric implants of ketoprofen, as well as the release of its pure S-isomer. The release profiles of rac-KET implants with 4%, 7%, and 10% drug loading in phosphate buffer were studied. The initial amounts of S-(+) and R-(−)-KET were the same because a mixture of racemates was used. In the first 28 days, the release of S-(+)-KET and R-(−)-KET were almost similar in all formulations with different concentrations, with large initial burst release in the first 10 h and a slow sustained release rate thereafter. However, after 42 days, a greater release of S-(+)-KET compared to R-(−)-KET began to occur. The differences between the two became more obvious as they progressed towards the end of the in vitro study. Although the S/R ratio diverged slightly from 1 at the beginning of the study, significant changes in favour of the S-enantiomer for all implants were observed after day 42. It is notable that the higher the drug load, the greater the deviation was [53].

However, reports on the effect of PLGA and other AIs on degradation parameters also appear in the literature.

Quan P. et al., in their recent study [54], demonstrated the effect of AI (donepezil hydrochloride) on PLGA systems, dominating over the above-discussed factors of the co-polymer itself—end groups, molecular weight, etc.—as well as other factors of the co-polymer itself. PLGA 5065 (lactide/glycolide = 50/50, Mw = 40,000 Da, with acid end groups), PLGA 5065E (lactide/glycolide = 50/50, Mw = 40,000 Da, acid end groups), and PLGA 5085E (lactide/glycolide = 50/50, Mw = 70,000 Da, end groups) were selected for the study. Gel permeation chromatography showed that the degradation rate of PLGA was accelerated when donepezil hydrochloride was incorporated into the microspheres, and the molecular weight of all three PLGA species dropped abruptly to about 11,000 Da within the first three days. The main AI-induced catalysis effect may be responsible for the accelerated degradation of PLGA, which resulted in similar in vitro dopenesim hydrochloride release profiles from different PLGA matrices.

### 3.7. Polymer Concentration

In Santhosh Kumar J., different PLGA-based formulations were prepared: polymer in amounts of 300, 350, and 400 mg with constant 50 mg cytabarine and 800 mg DMSO as solvent were taken.

Analysis of those formulations via UV spectroscopy revealed a pattern: as the ratio of mass fraction of polymer to active pharmaceutical substance in the dosage form increases, the encapsulated drug content increases too. The most effective matrix loading was observed in the formulation with the highest polymer content. It was also observed that as the concentration of polymer decreased, porosity of the structure was increased, and more intense release was observed. The matrices with 33.3% polymer content showed prolonged release of the drug for 28 days. Marked burst release was observed in case of compositions with a polymer concentration of less than 31% by weight. For compositions with a PLGA content of 33.3%, only 20–25% burst release was observed [55].

In the work of Bode C. et al., where the mechanism of release from PLGA-based implants was studied, colourants with different affinity to water were added to the polymer matrix. During the implants’ manufacturing, *N*-methylpyrrolidone (NMP) was used as a solvent and the environment was a phosphate buffer with pH 7.4, from which water, by phase inversion and simple diffusion, passed into the polymer, while NMP passed into the aqueous volume of the solution. Thus, at the border of the compound–water phase, the polymer solubility gradually decreased and a shell was formed, which grew towards the implant core. The hollow central part was firstly filled with NMP/water, then more with water. In the initial composition containing 45% PLGA, the implant “walls” were thicker compared to the one with 30% PLGA. It was experimentally shown that the rate of water penetration through the thicker “walls” was reduced, as was the release of methylene blue (water soluble). Earlier, it was found that thicker implant walls, as a result of higher polymer concentrations, also lead to more significant autocatalytic effects and accelerated degradation of PLGA [56].

### 3.8. Polymer Loading with Active Ingredient

Usually, the loading of PLGA-based matrices with the active substance does not exceed 35%. With increasing percentage of drug concentration in ISFI, the water absorption and swelling processes described earlier alter. According to the study, a strong correlation between the total mass loss of the implant and the amount of drug released was observed. The method of sampling the whole implant was used to determine this dependence. This trend indicates the possibility of a drug release mechanism based on the erosion process, in which the polymer and the drug are released at the same rate.

In the experiment described by Costello MA et al., it was observed that dexamethasone interacts limitedly with the PLGA polymer in the implant, which reduces the ability of AI to penetrate through the polymer matrix. It is assumed that the release of dexamethasone in this case occurs mainly due to the limited access of water to its crystals. The release of the active substance from the implant is possible only after PLGA breaks down to the degree that allows oligomers and monomers to dissolve and thus provides availability to a certain amount of water for dexamethasone dissolution [57].

Budhian A. et al. considered concentrations of 0.66, 1.7, and 2% haloperidol loaded in PLGA microparticles, and evaluated the effect of this on the “burst” release factor [58]. It was shown that the profile of change in the percentage of release may not depend significantly on AI release concentration, but the absolute value of haloperidol released changes equivalently with increasing loading. For example, for PLGA systems containing 0.66%, it was 7 mg/mL during the first hours of release, while for microparticles with 2% haloperidol, it was about 17 mg/mL during the same time.

An increase in the concentration of AI in the system means an increase in the number of particles both on the surface of the implant and in its core. The particles in the surface layers of the implant will be responsible for the “burst” release effect, whereas the particles in the core of the system will be responsible for the remaining release time.

### 3.9. Combination of Composition

PLGA is known to retain a relatively hard texture under a wide range of biological conditions due to its glass transition temperature, which is typically above 37 °C. This is an important factor for ISFIs of various applications—especially those implanted in the tooth socket after tooth extraction. As mentioned above, the solidity of a PLGA-based system is influenced by the amount of LA content. However, it is not reasonable to create a system for implantation of the required firmness only by varying the LA:GA ratio, as the change in the fragments ratio will affect many other parameters of the system. To improve matrix properties such as solubility, viscosity, and glass transition temperature, combinations of different matrix-forming polymers are widely used. As the research shows, it is also possible to influence AI release parameters using such combinations.

In a study by Karp F. et al., a combination matrix, which is a blend of PLGA and Eudragit^®^ polymethacrylic polymer, was studied for delivery and modified release of florfenicol. The ionic polymer used in the study was Eudragit^®^ E100, which is a cationic polymer, and Eudragit^®^ S100 polymer, which is an anionic polymer, with Eudragit^®^ E100 dissolving well at pH 5 and Eudragit^®^ S100 dissolving well at pH 7.6. It was shown that the release rate of florfenicol was higher with the cationic Eudragit^®^ E100. This may be due to the pH microenvironment created during degradation of the PLGA polymer. The researchers hypothesised that the solubility of Eudragit^®^ E100 increases at low pH values, leading to destabilisation of the polymer implants. On the other hand, Eudragit^®^ S100 loses its solubility in an acidic environment, maintaining the composition of the polymer matrix [59].

Cao Z. et al. developed an injectable depot system of PLGA and SAIB (sucrose acetate isobutyrate) for progesterone delivery [60]. It was shown to effectively reduce the administered dose and reduce the degree of “burst” release when using the combined matrix.

Several works are also devoted to the combination of β-cyclodextrin and PLGA matrices [61,62]. As a result of the addition of this co-matrix-former in the study of Zheng K. et al. [62], it was possible to slow down the release of paclitaxel from nanoparticles, as well as to significantly increase the bioavailability—the AUC value for the combined matrices was 2.4 times higher than that of the commercial drug Taxol^®^, and 1.7 times higher than that of conventional PLGA nanoparticles.

Additional risks should also be considered when co-matrix formulations are used. In a study by Duque L. et al., shellac was used as an acid-resistant co-matrix former for a PLGA-based implant delivering a highly sensitive molecule of protein nature [63]. Along with successful protection of acid-labile protein AI, shellac in the ratio from 10:1 to 1:1 to PLGA in the implant composition increased the value of “burst” AI release by more than 4 times, which, in the authors’ opinion, can be related to its partial dissolution in a neutral environment.

### 3.10. Use of Cosolvents

The study of the effect of dissolving ability and different solvents’ plasticising effects on the gelation and solidification rate of the implant is an important aspect in achieving the desired drug release profiles. The use of water-miscible (hydrophilic solvents) and immiscible (hydrophobic solvents) substances is considered the best strategy to create slow drug release profiles from ISFI, while ensuring suitable viscosity and phase inversion. Solvents such as triacetin and ethyl acetate are poorly miscible with water and can contribute to a slower and more sustained drug release profile compared to ISFI systems containing a solvent such as NMP.

This happens due to the rapid extraction of the hydrophilic solvent, which mixes with water, leading to a rapid conversion of the solution into a gel and then to the formulation of a dense matrix. On the other hand, the hydrophobic triacetin does not show ISFIs gel transition immediately after the introduction into the liquid environment due to less miscibility and slower diffusion of the solvent into the aqueous phase. The use of NMP as a solvent can lead to the formation of ISFI with an irregular porous structure. This is due to rapid exchange between NMP and aqueous phase, which enhances the drug release. The use of triacetin leads to the formation of a system with a less porous structure and slower phase inversion, resulting in a compact and dense ISFI structure and slower AI release. Implants containing 40% PLGA-triacetin with low solubilisation potential of AI release showed, in a recent study by Gomaa E. et al., a more sustained cumulative drug release than those containing solvents with higher solubilisation (ethyl acetate and NMP), which was 93.06% after 21 days. This formulation was chosen for further study because of the low burst release profile demonstrated with triacetin and the optimally high PLGA concentration [64].

Liu and Venkatraman [65] also reported the slow release of metoclopramide monohydrochloride, which was observed when hydrophobic triacetin was added as a co-solvent to the hydrophilic solvents used (NMP and DMSO). The suppression of burst release of the drug was explained by slow solvent exchange and delayed phase inversion. In another study, Liu et al. [66] studied the effect of triacetin in achieving lower initial release and continuous release of thymosin alpha-1 up to one month in vivo due to slower polymer phase inversion and denser structure of solidified ISFI systems formed using such hydrophobic solvents [64].

Various vegetable oils are also used as hydrophobic solvents in phase-sensitive in situ systems. Numerous studies by Thawatchai Phaechamud et al. have been dedicated to the study of the effect of peppermint oil, clove oil, and lime oil on the release of various active ingredients [67]. In fact, by adjusting the amount of oil added, it is possible to change the polymer properties such as viscosity, water diffusion rate, pH, wettability, injectability, etc., which, in turn, help in reducing the drug release [68].

### 3.11. Inclusion of Excipients

Another way to influence the physicochemical properties of the polymer-based system is the addition of auxiliary substances (AS). In general, these excipients can be divided into three main categories with respect to their affinity for water: hydrophilic, hydrophobic, and amphiphilic [69]. To accelerate the release of active ingredients, substances such as mannitol [70] and polyvinylpyrrolidone [71] should be added to the polymer-based system to enhance the interface and diffusion. Hydrophilic ASs can affect the pH, altering it to an alkaline side, which, during the initial release, can lead to morphological changes [70].

The introduction of hydrophobic excipients can reduce the rate of drug release, create less porous sponge-like matrices, and lead to reduced initial breakdown of a polymer and prolonged drug release by retaining a large amount of solvent. Examples of such additives are stearic acid, glycerol monostearate, methyl heptanoate, ethyl heptanoate, and ethylnonanoate [72,73,74]. Moreover, it is known that some hydrophobic excipients (for example, lipid-based nanoparticles) can increase bioavailability upon exposure and use on mucous membranes and intact skin [75,76].

The presence of a hydrophilic polymer such as poloxamer 188 or PEO tended to aid water penetration into these structures, limiting PLGA degradation and drug release through cleavage of the ester linkage. This is because the mobility of the acids formed (and bases coming from the surrounding fluid) is increased in these implants. Consequently, acids are neutralised faster and micro-pH variations are less expressed. Thus, autocatalytic effects are less critical and PLGA degradation is slower in implants containing poloxamer/PEO compared to implants containing pure PLGA-AI [77].

Amphiphilic AS such as Pluronic^®^VR (poly(ethylene)oxide-poly(propylene)oxide-poly(propylene)oxide) (PEO-PPO-PEO) [48,69] also affect the polymer morphology. The proper choice of concentration and chain length allows the polymer to achieve a hydrophilic–lipophilic balance, which results in a higher water uptake due to the presence of hydrophilic PEO ends and the diffusion barrier carried out by the hydrophobic PPO chains [78]. In addition, Pluronic^®^VR can contribute to the separation in the phase inversion process corresponding to the specific arrangement of its chains in the system. The hydrophobic PPO chains are embedded in the polymer matrix, resulting in reduced adsorption of AIs and increased biocompatibility of the system. Simultaneously, the hydrophilic ends of PEO diffuse into the external environment, creating a diffusion barrier.

### 3.12. Release Environment

In the process of AI release from polymer matrices into the surrounding biological fluid, the main external factor that affects their release rate is the acidity/basicity of the environment. The pH affects the degradation speed of the polymer and the stability of the active molecule. PLGA studies have shown that at pH 2.4, the polymer surface is smooth with no pores and undergoes homogeneous biodegradation. However, when the pH is altered to an alkaline environment, such as at pH 7.4, the polymer has a porous structure and degrades heterogeneously, and the release rate of the active ingredient decreases tremendously. An increase in the ionic strength of the environment can also lead to a slower release of the therapeutic agent. This is because high ionic strength can reduce the degree of swelling of the polymeric matrix containing the drug. Swelling of the polymer occurs due to the permeation of the environment in the presence of ions, which increases the polymer matrix volume. Reduced diffusion of the encapsulated drug compound may also contribute to delayed release [79].

Another factor that can affect release is temperature. For example, if it is raised, the mobility of the polymer molecules may increase, which in turn leads to its extension and pore closure [80]. If the surface layer of the matrix remains intact, it may slow down the drug release. At the same time, in vivo studies have shown faster release and shorter lag phase of degradation when temperature is increased, due to changes in enzyme activity.

Plasticising agents, salts, and surfactants in the liquid phase can also influence drug release processes. However, in contrast to encapsulation of active components, high osmolality of the release environment can slow down the diffusion of AIs by reducing the rate of water absorption by the polymer [81].

It should be noted that even though, in this strategy, factor-12 is classified as a variable factor to be changed in the process of testing, it is the only one among all the factors that is exogenous and can be easily adjusted only within the in vitro studies.

## 4. Discussion

Drug release from delivery systems in vitro (cumulative release) can be described by one of the types schematically imagined in Figure 5 [9].

The number of cumulative release phases typically varies from one to three. When the in vitro release profile is monophasic (Figure 5A), the zero-order kinetics equation is suitable for describing the release, in which the same AI release dose is released from the system at each time point. This type of release is called controlled release. Biphasic release (Figure 5B,C) may be characterised by an “burst” release at the first moments of time (from several hours to several days) and then a plateau (“power-law phase”). In another case, biphasic release does not have the effect of an initial “burst” but is characterised by a lag phase and a transition to a “power-law phase”.

The triphasic release (Figure 5D), in contrast to the biphasic release with a “burst” effect (Figure 5B), has a third phase of “accelerated release” characterised by polymer degradation and a sharp increase in the amount of AI release. To achieve a triphasic release, a moderate “burst” effect should be maintained, but the proportion of AI released during this phase should be small relative to the total load.

Obviously, it cannot be argued that “burst” release is always negative for the delivery system being developed and must be adjusted in all cases. On the contrary, for modern antimicrobial drugs, antiretrovirals, and some other agents in delivery systems for long-term implantation, scientists are trying to achieve the release of an “impact” dose during the first hours or days, with a gradual release of the residual dose in the future. In particular, this is due to the need to stop the formation of a bacterial biofilm or to compensate for the inflammatory process after invasion [82,83].

Thus, in a recent study by He X. et al. [84], an optimal system for the delivery of gonadotropin-releasing hormone agonists, whose satisfactory release profile is a high initial release followed by a small amount of drug release per day, was developed.

In this work, NaCl, CaCl_2_, and glucose were selected to improve the release profile of model AI from PLGA microspheres. The efficiency of pore formation in the polymer matrix was similar with each of the three additives. The effect of the three additives on drug release was evaluated. At optimum initial porosity, the initial number of released microspheres containing different additives was comparable, providing a good inhibitory effect on testosterone secretion at an early stage. However, it was found that the addition of microspheres with glucose could not only increase the initial release of the drug but also promote subsequent controlled release. The reason why the incorporation of glucose delayed the subsequent release of the drug was investigated. SEM results showed that during the formation of glucose containing microspheres, a significant number of pores are closed. Further, thermal analysis was carried out and a clear decrease in glass transition temperature was observed in this composition. During the decrease in glass transition temperature, the polymer chains can rearrange at lower temperatures. This change in properties was reflected in the gradual closure of pores and was probably responsible for the delayed release of the drug substance after initial release [84].

In a study by Du C. et al., the design of the experiment to create a new antimicrobial agent for chronic recurrent bacterial infections determined the need for both short-term action of the drug (due to “burst release in the first 18 h”) and long-term action over 7 days [85].

Also, in the study by Jain P. et al. [86] on the periodontal delivery of microspheres based on PLGA and chitosan loaded with doxycycline hyclate, that composition was substantiated (using the QbD approach) as optimal, which demonstrated a biphasic release: 25.95% of doxycycline hyclate was released in the first 16 h of the in vitro dissolution test, with the second phase demonstrating sustained release over 14 days. This kind of research can significantly advance modern medicine in solving a socially significant problem—the effective treatment of chronic periodontitis, through long-term installation of medicine in periodontal pockets and interdental spaces [87].

## 5. Conclusions

Creation of in situ formed implants based on PLGA is a promising direction of modern pharmaceutical developments. But for successful completion of the development cycle of such systems, it is necessary to solve the problem of “burst” release characteristic for this type of matrix. For this purpose, a systematic approach called “System of 12 factors” was proposed, including constant and variable, endogenous and exogenous factors that can influence the severity of burst AI release from PLGA matrix. It is important to note that to create an optimal composition for implantation, it is sometimes necessary not to completely eliminate the phenomenon of “burst” release but to adjust the amount of the substance released in the first hours or days for the transition of the release profile into a triphasic state.

## Figures and Tables

**Figure 1 pharmaceutics-16-00115-f001:**
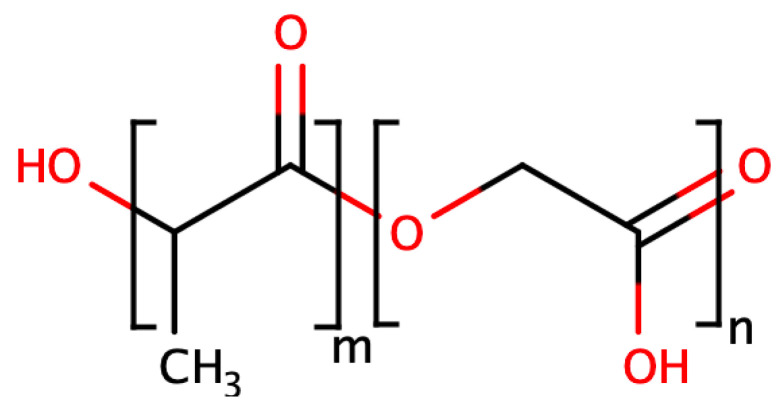
PLGA block copolymer formula: m-number of lactide fragments in the PLGA chain; n-number of glycolide fragments in the PLGA chain.

**Figure 3 pharmaceutics-16-00115-f003:**
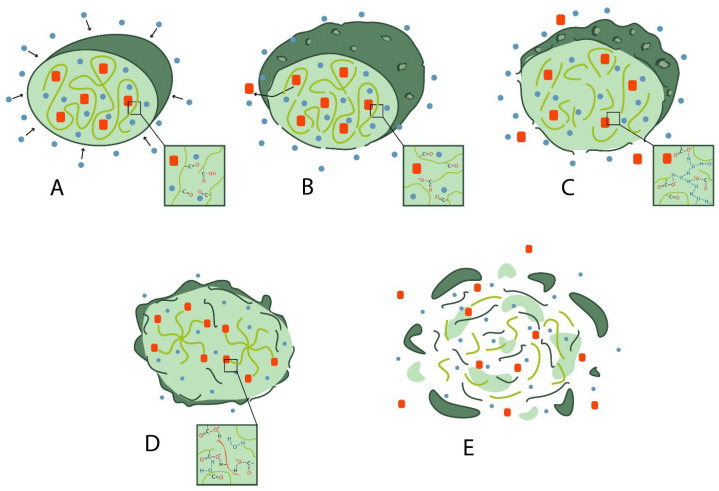
Active ingredient release from PLGA-based matrices: (**A**) diffusion of fluid from the surrounding to the inside of the matrix; (**B**) polymer swelling and pore formation; (**C**) degradation and erosion; (**D**) drug–polymer interaction; (**E**) complete polymer degradation.

**Figure 4 pharmaceutics-16-00115-f004:**
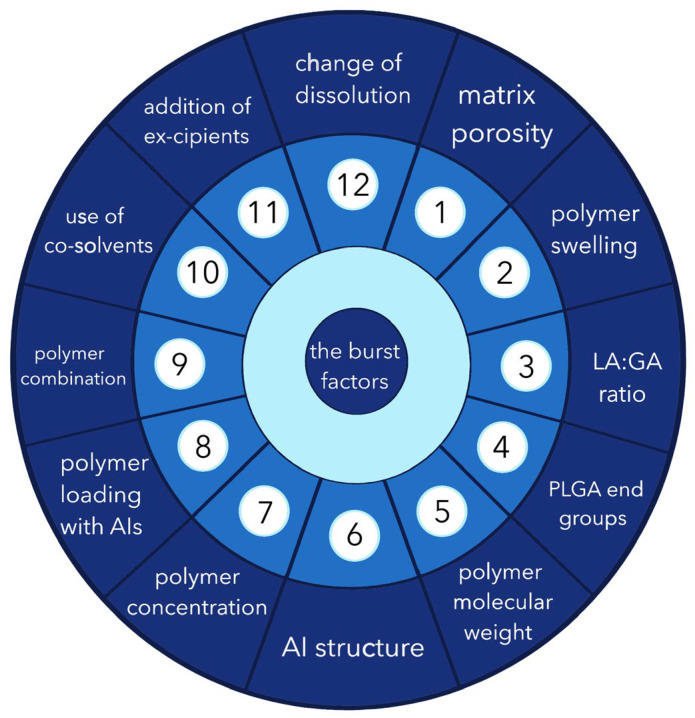
Twelve effectors of the AIs “burst” release from PLGA matrices: 1—matrix porosity; 2—polymer swelling; 3—LA:GA ratio; 4—PLGA end groups; 5—polymer molecular weight; 6—AI structure; 7—polymer concentration; 8—polymer loading with AIs; 9—polymer combination; 10—use of co-solvents; 11—addition of excipients; 12—change of dissolution conditions.

**Figure 5 pharmaceutics-16-00115-f005:**
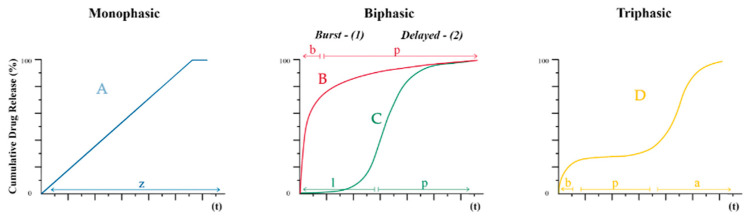
Types of cumulative AI release: A—monophasic; B—biphasic with “burst” release effect; C—biphasic delayed; D—triphasic. Four “phases” of drug release: z—zero-order release phase; b—burst release phase; p—power-law phase; l—lag phase; a—accelerated release phase.

## Data Availability

The data presented in this study are available in this article.

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
