# Peer review of "Burst Release from In Situ Forming PLGA-Based Implants: 12 Effectors and Ways of Correction"

_pharmaceutics, 2024, doi:10.3390/pharmaceutics16010115_

Round 1
Reviewer 1 Report
Comments and Suggestions for Authors
Dear authors,
The manuscript is very well written and presents scientific and main issues. I would suggest some minor changes shown below:
- What does “AI” stand for in the abstract? Please specify at the beginning of the text.
- Please provide a citation for the statement "The exception to this rule is the 50:50 copolymer, which degrades faster" in line 11. Also, please cite a refrence for the information in line 116.
• In line 123, please replace "and" with "of".
• In Figure 4, I suggest you include the text "Twelve effectors of the AIs" in the figure itself, as it will have a greater impact on the reader.
• In line 204, please elaborate on the sentence "increasing the mobility of diffusing molecules can also help to prevent autocatalytic effects". Is there any reference to support this claim?
• In line 212, you wrote "The results showed that the release rate of AI was almost unchanged with increasing microparticle size and decreasing drug concentration gradient." Considering the effect of AI, it would be better to mention the name of AI in this case and in other cases where you compare it with other studies, such as Yuan P. et al. (2019). Please specify which AI you are referring to.
• In line 244, what do you mean by "particles" in this sentence? Please clarify.
• In line 268, please indicate the dimension of "p=0.048".
• In lines 262 and 269, there seems to be a mismatch between the effect of lactide content and crystallinity. Please check and correct this inconsistency.
• In the section "3.7. Polymer concentration", the first paragraph contradicts the second paragraph. Please explain this discrepancy and ensure it is consistent with the main article.
• In some cases, depending on the type of drug and its mechanism of action, creating a burst release of the drug can have a positive effect on its performance. For example, in antibiotics, creating an initial burst release of the drug creates a loading dose, which leads to a greater effect of the antibiotic Please refer to this aspect as well.
Regards,
Author Response
Dear Reviewers,
Hello! Our team of authors would like to thank the highly respected reviewers for their extensive work in correcting the manuscript and giving us constructive suggestions for it improvement.
During this round of post-review revisions, the authors revised the title, abstract and keywords of the manuscript. Figure 4 was updated and 7 literature sources were added, including those recommended by the esteemed reviewers.
Also we have clarified all the abbreviations used in our manusript, changed some prepositions, checked out English and tried to eliminate all the contradictions in our work.
About p-factor for which you asked for its dimension — this factor doesn’t have any dimensions because it is just a mathematical (statistical) factor which shows us the probability of obtaining results at least as extreme as the observed results of a statistical hypothesis test, assuming that the null hypothesis is correct (according to definition).
We hope to hear from you soon!
Best regards,
PhD student,
Victor Pyzhov

Reviewer 2 Report
Comments and Suggestions for Authors
1. Revise English language structural and grammatical
2. use the following recent published reviews to explain the basics of the 11 system for example journal of controlled release ( 362, 70-96, 2023) and international journal of biological macromolecules ( 127672, 2023) Amon others.
3. Restructure the abstract
4. modify the title to reflect the review content
Comments on the Quality of English LanguageAcceptable with minor corrections
Author Response
Dear Reviewers,
Hello! Our team of authors would like to thank the highly respected reviewers for their extensive work in correcting the manuscript and giving us constructive suggestions for it improvement.
During this round of post-review revisions, the authors revised the title, abstract and keywords of the manuscript. Figure 4 was updated and 7 literature sources were added, including those recommended by the esteemed reviewers.
All comments on improving English language and phrase construction were accepted and corrections were made. Abbreviations were also clarified and additional information from the primary sources of the articles requested by the reviewers was provided.
The changes made are highlighted in the text in colour.
With best regards,
PhD student,
Victor Pyzhov

Reviewer 3 Report
Comments and Suggestions for Authors
Dear Authors,
you made a great work! However, some improvements are mandatory before acceptance.
The paper is a review on the burst release from in situ forming PLGA-based implants. The Authors made a great work in terms of methodology and the paper sounds scientific and well written. However, some improvements are mandatory before acceptance.
The abstract is well written, complete and summary in its various aspects.
The keywords are complete and appropriate.
The Introduction is well written, clear and complete in many respects. I think the authors have done a great job analyzing this topic with such great clarity.
Section 2 “Base PLGA characteristics” it is absolutely well written and clear from several points of view, capable of making the reader aware of the characteristics of this product.
Section 3 “Possible causes of the "burst release" problem” it is relevant to the topic covered, and inserts useful general information considering the specific analysis of the topic that is proposed immediately afterwards:
- Porosity
- Swelling of the polymer
- LA/GA ratio
- Impact of end groups
- Molecular weight of the polymer
- Structure of the active molecule
- Polymer concentration
- Polymer loading with active ingredient
- Combination of composition
- Use of cosolvents
- Inclusion of excipients
- Release environment
The different sections into which the manuscript has been divided represent it in a precise and accurate manner. I suggest summarizing these sections as much as possible, the content is adequate but the form is verbose.
I think it might be interesting to include some information regarding the selection of the Articles, how and which ones were considered in the study and what information is reported from them?
In the discussion:
- As indicated in the introduction, similar technology can also be used in Dentistry. From this point of view, it is interesting to consider the possibility that it is useful at a global action level in a district that is difficult to reach for a long time and in several particularly hidden areas, such as periodontal pockets or interdental spaces, as indicated by: “Mahendra J, Mahendra L, Mugri MH, Sayed ME, Bhandi S, Alshahrani RT, Balaji TM, Varadarajan S, Tanneeru S, P ANR, Srinivasan S, Reda R, Testarelli L, Patil S. Role of Periodontal Bacteria, Viruses, and Placental mir155 in Chronic Periodontitis and Preeclampsia-A Genetic Microbiological Study. Curr Issues Mol Biol. 2021 Jul 29;43(2):831-844. doi: 10.3390/cimb43020060.”
Conclusions are concise and clear. Bibliography should be formatted respecting the journal’s requirements and no improper citations are evidenced. Figures and labels are clear and easy to comprehend. English is clear and easy to understand.

Author Response
Dear Reviewer,
Hello! Our team of authors would like to thank the highly respected reviewers for their extensive work in correcting the manuscript and giving us constructive suggestions for it improvement.
During this round of post-review revisions, the authors revised the title, abstract and keywords of the manuscript. Figure 4 was updated and 7 literature sources were added, including those recommended by the esteemed reviewers.
All comments on improving English language and phrase construction were accepted and corrections were made. Abbreviations were also clarified and additional information from the primary sources of the articles requested by the reviewers was provided.
The changes made are highlighted in the text in colour.
We hope to hear from you soon!
Best regards,
PhD student,
Victor Pyzhov
